# Systematic review and individual-patient-data meta-analysis of non-invasive fibrosis markers for chronic hepatitis B in Africa

Asgeir Johannessen [1,2,23] ✉, Alexander J. Stockdale [3,4,23],
Marc Y. R. Henrion [4,5,23], Edith Okeke[6], Moussa Seydi[7], Gilles Wandeler[8],
Mark Sonderup[9], C. Wendy Spearman [9], Michael Vinikoor[10,11], Edford Sinkala[10],
Hailemichael Desalegn [1,12], Fatou Fall[13], Nicholas Riches[5], Pantong Davwar[6],
Mary Duguru[6], Tongai Maponga [14], Jantjie Taljaard[15],
Philippa C. Matthews[16,17,18], Monique Andersson [14,16], Souleyman Mboup[19],
Roger Sombie[20], Yusuke Shimakawa [21,24] & Maud Lemoine[22,24]

In sub-Saharan Africa, simple biomarkers of liver fibrosis are needed to scale-up hepatitis B treatment. We conducted an individual participant data meta-analysis of 3,548 chronic hepatitis B patients living in eight sub-Saharan African countries to assess the World Health Organization-recommended aspartate aminotransferase-to-platelet ratio index and two other fibrosis biomarkers using a Bayesian bivariate model. Transient elastography was used as a reference test with liver stiffness measurement thresholds at 7.9 and 12.2kPa indicating significant fibrosis and cirrhosis, respectively. At the World Health Organization-recommended cirrhosis threshold (>2.0), aspartate aminotransferase-to-platelet ratio index had sensitivity (95% credible interval) of only 16.5% (12.5–20.5). We identified an optimised aspartate aminotransferase-to-platelet ratio index rule-in threshold (>0.65) for liver stiffness measurement >12.2kPa with sensitivity and specificity of 56.2% (50.5–62.2) and 90.0% (89.0–91.0), and an optimised rule-out threshold (<0.36) with sensitivity and specificity of 80.6% (76.1–85.1) and 64.3% (62.8–65.8). Here we show that the World Health Organization-recommended aspartate aminotransferase-to-platelet ratio index threshold is inappropriately high in sub-Saharan Africa; improved rule-in and rule-out thresholds can optimise treatment recommendations in this setting.

Worldwide, an estimated 316 million people live with chronic hepatitis B virus infection (CHB)[1]. The natural course of infection is variable, ranging from an inactive carrier with an excellent long-term prognosis, to progressive hepatic necroinflammation leading to cirrhosis and/or hepatocellular carcinoma (HCC)[2]. Antiviral therapy effectively reduces the risk of these complications[3,4], and the challenge in clinical practice is to identify patients at risk of progressive liver disease who should start timely antiviral therapy.

International treatment guidelines recommend antiviral therapy for patients with: cirrhosis; those with elevated hepatitis B virus (HBV) viral load and significant liver fibrosis; or those with high viraemia and inflammation[5–8]. Cirrhosis can be diagnosed clinically mostly at an

advanced, decompensated phase. Earlier stages of liver fibrosis have traditionally been assessed by liver biopsy, which recently has been largely replaced by transient elastography (TE)[9]. In resource-limited settings, however, these fibrosis assessment tools are rarely available, and antiviral treatment is therefore often delayed until the patients have developed symptoms of advanced chronic liver disease (CLD).

The first World Health Organization (WHO)'s guidelines for CHB published in 2015 recommended the use of non-invasive fibrosis markers based on low-cost routinely available laboratory tests in resource-limited settings, and this was adapted into national guidelines in many low- and middle-income countries. Specifically, the aspartate aminotransferase-to-platelet ratio index (APRI) at a threshold of 2.0 was recommended to identify patients with cirrhosis, although the lack of data from sub-Saharan Africa (sSA) was acknowledged as an important knowledge gap[8]. APRI, FIB-4 and most other liver fibrosis markers were developed and validated in Caucasian and Asian cohorts[10,11], where environmental exposures, endemic infections, and host/viral genetic factors differ from sSA.

With an estimated 82 million people living with CHB, sSA is in desperate need of locally-adapted treatment guidelines for CHB that consider operational constraints and potential differences in the natural history of infection in the region[12]. To meet this demand, we conducted a systematic review and individual patient data (IPD) meta-analysis to evaluate the performance of APRI, FIB-4, and gamma-glutamyl transferase-to-platelet ratio (GPR), a simple liver fibrosis biomarker developed in West Africa[13], for the diagnosis of significant fibrosis and cirrhosis in CHB patients living in sSA. Although liver biopsy is considered the gold standard to diagnose these conditions, we used TE as a reference test in this analysis since liver biopsy is rarely performed in sSA and TE is now widely recognised as a reliable alternative to liver histology. TE has been recommended as a decision tool for HBV treatment in international HBV guidelines, and liver stiffness measured by TE predicts the risk of hepatic decompensation, incident HCC, oesophageal varices, and mortality[14]. The use of TE has been well validated compared to liver histopathology in several studies of CHB patients in sSA[13,15,16].

## Results
### Study population
Database searches identified 1478 articles following removal of duplicates. After screening of title and abstract, 90 potentially eligible articles were fully reviewed (Supplementary Fig. 1). Finally, 30 articles met our inclusion criteria, and all the authors of these articles agreed to share the IPD[13,15–43]. We additionally included IPD data from yet unpublished cohorts from South Africa (Spearman & Sonderup, 2021) and Dakar, Senegal (Seydi & Wandeler, 2021), and from one cohort which has subsequently been published (Thiès, Senegal)[44].

Overall, we obtained IPD of 3960 hepatitis B surface-antigen (HBsAg)-positive patients from 12 distinct cohorts in eight countries: Burkina Faso, Ethiopia, The Gambia, Malawi, Nigeria, Senegal, South Africa, and Zambia (Fig. 1). We excluded 412 ineligible participants (Fig. 2). Characteristics of individual cohorts are reported in Supplementary Table 1.

Assessment of study quality indicated an overall low risk of bias according to QUADAS-2 criteria (Supplementary Table 2). All studies systematically performed both index tests and reference tests, and >90% of study participants had blood tests and TE the same week. Two cohorts had specific criteria that could reduce applicability: one restricted to asymptomatic patients with HBV DNA > 3.2 log10 IU/ml[16], and another excluded patients with body mass index (BMI) > 28 kg/m$^2$ [23]. Three studies performed TE in a subset of the overall cohort with non-random selection: two due to equipment availability[15,36], and one at clinician's discretion without specifying criteria[30]. Two studies had significant loss to follow-up between community diagnosis and evaluation at clinic[37,38].

Table 1 summarises characteristics of the 3548 study participants. Median age was 33 years (interquartile range [IQR] 28–41), 60% were male, and median BMI was 22.4 kg/m$^2$ (IQR 20.0–25.5). In >80% of cases (2824/3497), the reason for HBsAg testing was asymptomatic screening, as part of population-based screening, antenatal care, blood donation, or because of a family contact with HBV, whereas 673 individuals (19%) were tested due to suspected CLD, viz symptoms or clinical signs, or abnormal liver enzymes. The prevalence of liver stiffness measurement (LSM) > 12.2 kPa (LSM > 12.2; associated with cirrhosis) was 7.3% overall: 2.5% among asymptomatic screening participants and 26.4% among patients with suspected CLD. LSM > 12.2 was more prevalent among men relative to women (9.9 vs. 3.4%; $P < 0.001$) and its prevalence increased with age (Supplementary Fig. 2). LSM > 7.9 kPa (LSM > 7.9; associated with significant liver fibrosis (≥F2)) was observed in 17.4% overall; 11.5% among asymptomatic screening participants and 40.7% among patients with suspected CLD ($P < 0.001$).

In a multivariable model, increased likelihood of LSM > 12.2 was observed with increasing age (per 1-year increment; Adjusted Odds Ratio [AOR] 1.03; 95% CI 1.01–1.04), male sex (AOR 3.28; 95% CI 2.17–4.96), and suspected CLD (vs. asymptomatic screening participants; AOR 55.3; 95% CI 27.9–109.3). Male sex and suspected CLD were also significantly associated with LSM > 7.9 (Supplementary Table 3).

### Diagnostic performance of APRI, GPR and FIB-4
APRI and GPR had the best discriminant performance, with area under the receiver operating curve (AUROC) of 0.81 (95% credible interval (CrI) 0.81–0.82) and 0.82 (95% CrI 0.81–0.83) for LSM > 12.2 and 0.75 (95% CrI 0.74–0.75) and 0.76 (95% CrI 0.75–0.77) for LSM > 7.9, respectively (Supplementary Fig. 3). FIB-4 had relatively lower AUROC of 0.77 (95% CrI 0.76–0.78) for LSM > 12.2 and 0.68 (95% CrI 0.68–0.69) for LSM > 7.9. ALT as a standalone marker was associated with the lowest performance, both for LSM > 7.9 and LSM > 12.2 (Fig. 3). Performance was significantly better for LSM > 12.2 relative to LSM > 7.9 for all evaluated biomarkers (Fig. 3).

By applying the WHO-recommended APRI thresholds (2.0 for cirrhosis and 1.5 for significant fibrosis), the sensitivity was 16.5% (95% CrI 12.5–20.5) for the diagnosis of LSM > 12.2 and 11.8% (95% CrI 9.4–14.2) for LSM > 7.9, whereas the specificity was 99.5% (95% CrI 99.2–99.7) for LSM > 12.2 and 99.2% (95% CrI 98.9–99.5) for LSM > 7.9.

We then developed rule-out thresholds aiming for a test sensitivity of ≥80% (Fig. 3). For APRI the optimised threshold to rule-out LSM > 12.2 was 0.36 with a sensitivity of 80.6% (95% CrI 76.1–85.1) and a specificity of 64.3% (95% CrI 62.8–65.8). For GPR the optimised rule-out threshold for LSM > 12.2 was 0.23 with a sensitivity of 80.6% (95% CrI 75.1–86.2) and a specificity of 66.7% (95% CrI 64.1–69.5).

The optimised rule-in thresholds for LSM > 12.2, where specificity exceeded 90%, were 0.65 for APRI with a sensitivity of 56.2% (95% CrI 50.5–62.2) and a specificity of 90.0% (95% CrI 89.0–91.0); and 0.47 for GPR with a sensitivity of 58.6% (95% CrI 50.7–66.3) and a specificity of 90.9% (95% CrI 88.7–92.9). We assessed the proposed cut-offs in each individual site (Supplementary Table 4). Significant inter-site heterogeneity was observed; however, due to a small number of patients with LSM > 12.2 at most sites, confidence intervals were wide at the level of individual centres.

As expected, predictive values were strongly associated with disease prevalence (Table 2). In asymptomatic screening populations, using the new rule-in thresholds for APRI (0.65) and GPR (0.47), positive predictive values for LSM > 12.2 were 17.0% for APRI and 20.1% for GPR, while negative predictive values were 98.4% and 98.5%, respectively. Among patients with suspected CLD, the positive predictive values using the same thresholds were 70.5% and 74.9%, whereas negative predictive values were 77.3% and 73.0%, respectively. The trade-off between optimising test sensitivity and specificity is

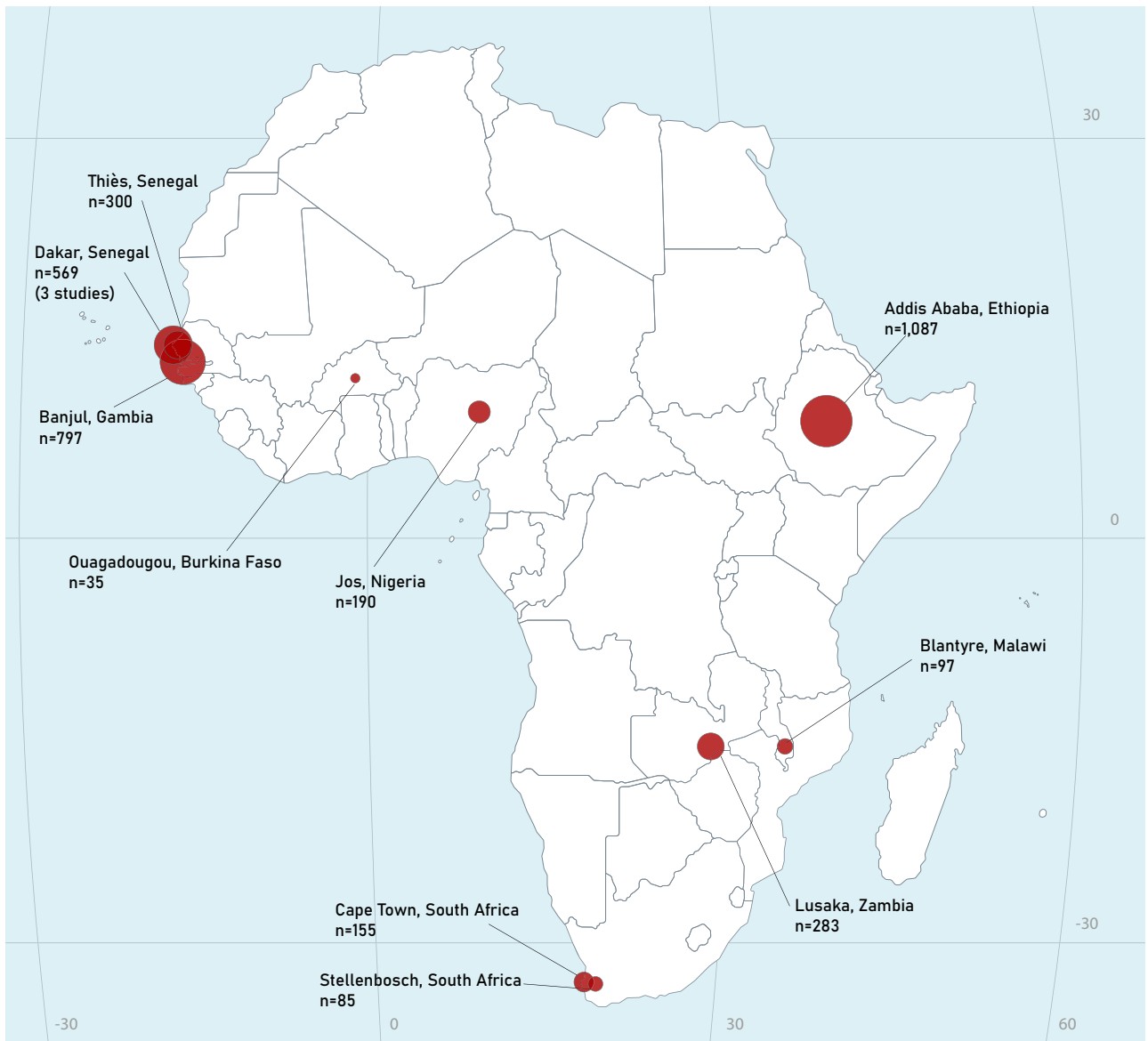

**Fig. 1 | Map of included studies.** n the number of hepatitis B patients included from each site in the current analysis. Area of circles is proportionate to cohort size. Attribution to iStockphoto (https://www.istockphoto.com/vector/africa-map-gm517581017-49316640).

illustrated in Fig. 4. Lower biomarker thresholds were associated with substantial over-diagnosis of LSM > 12.2 (Fig. 5).

The following patient level co-variates were significantly associated with model performance for the APRI and GPR rule-in thresholds: hazardous alcohol consumption, suspected CLD as the reason for HBsAg testing, and female sex (Supplementary Table 5). Hazardous alcohol consumption reduced test specificity for both biomarkers. Specificity was improved for APRI and GPR for women relative to men with no significant effect of sex on sensitivity. For the GPR model, being overweight reduced test specificity. The effect of subgroup characteristics on diagnostic sensitivity and specificity is illustrated in Supplementary Fig. 4.

We performed a sensitivity analysis using centre-specific upper limit of normal (ULN) thresholds; these ranged from 25 to 60 U/L for aspartate aminotransferase (AST) and 24 to 85 U/L for gamma-glutamyl transferase (GGT). This was associated with reduced diagnostic performance. We then considered the effect of using a lower LSM cut-off (9.5 kPa) for the diagnosis of suspected cirrhosis. At this threshold, relative to 12.2 kPa, for both APRI and GPR, test

performance was poorer, with lower sensitivity and specificity at rule-in and rule-out thresholds (Supplementary Fig. 5).

In a sensitivity analysis, among a subset of 134 patients who underwent pre-therapy evaluation with a liver biopsy, we assessed the diagnostic characteristics of APRI and GPR using METAVIR histological fibrosis scores as a reference test (Supplementary Table 6). Consistent with the findings from the main analysis, the WHO-recommended threshold of 2.0 was associated with a sensitivity of 11.1% (95% CI 0.3–48.2) and specificity of 99.2 (95% CI 95.6–100) for the diagnosis of cirrhosis (F4). The derived rule-in threshold of 0.65 for APRI was associated with a sensitivity of 100% (95% CI 66.4–100) and a specificity of 73.6% (95% CI 65.0–81.1), whereas the rule-out threshold was associated with a sensitivity of 100% (95% CI 66.4–100) and a specificity of 36.0% (95% CI 27.6–45.1). Similar findings, consistent with the main analysis were observed for GPR, and for the diagnosis of significant fibrosis (F2), as shown in Supplementary Table 6.

In an exploratory analysis, we compared patients with LSM > 12.2 who had false negative APRI classification results with those correctly classified to identify factors associated with impaired test sensitivity.

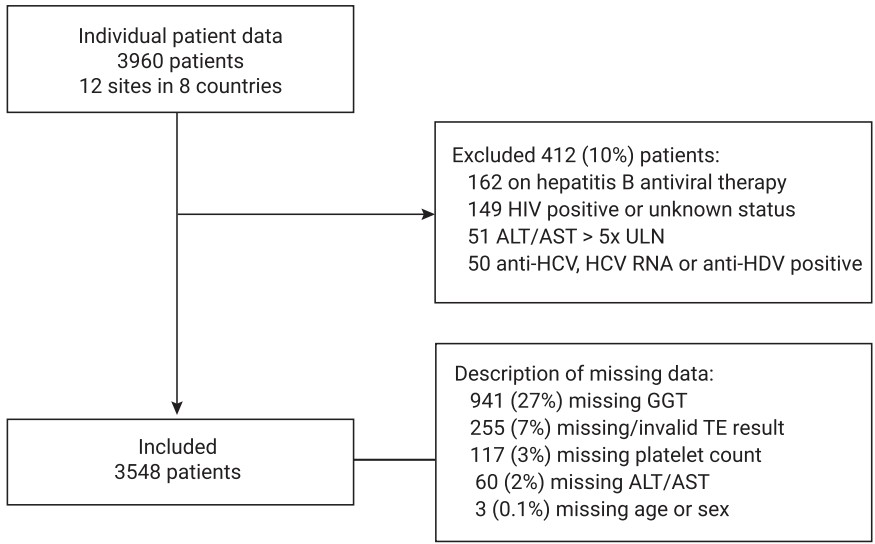

**Fig. 2 | Flowchart of data sources.** HIV human immunodeficiency virus, ALT alanine aminotransferase, AST aspartate aminotransferase, ULN upper limit of normal, HCV hepatitis C virus, HDV hepatitis D virus, GGT gamma-glutamyl transferase, TE transient elastography.

The lowest sensitivity for APRI was observed among patients with LSM just above the threshold (12.2 kPa) for cirrhosis, and diagnostic sensitivity increased with increasing LSM with the effect plateauing above 30 kPa (Supplementary Fig. 6).

## Discussion

In this study including more than 3500 patients living with CHB in sSA, we evaluated the diagnostic performance of three inexpensive, simple liver fibrosis biomarkers (APRI, GPR and FIB-4). Our analysis included all identified existing published data from sSA, together with novel data, offering the most comprehensive evaluation of the performance of these diagnostic tools in sSA to date. We found that the sensitivity of APRI to diagnose LSM >12.2 kPa (associated with cirrhosis) at the WHO-recommended threshold of 2.0 was only 16.5%, consistent with previous reports from the region[13,15,17]. Importantly, we also identified new rule-in and rule-out thresholds suitable for sSA. The best test characteristics were observed for APRI and GPR for the detection of LSM > 12.2; an APRI rule-in threshold of 0.65 yielded 56.2% sensitivity and 90.0% specificity, whereas a rule-out threshold of 0.36 yielded 80.6% sensitivity and 64.3% specificity.

Our findings compare well with a recent large IPD analysis of mainly Asian CHB patients using liver biopsy as a reference standard[45]. The authors found that the conventional APRI and FIB-4 thresholds were unsuitable for patient management and identified new, lower rule-out cirrhosis thresholds for both APRI (0.45) and FIB-4 (0.70), in line with our findings. Of note, both our study and the study by Sonneveld and colleagues applied a fixed ULN for AST. In a sensitivity analysis, we observed a poorer performance of APRI when using assay-specific ULN. It is likely that the variable performance of APRI in previous studies could be, at least partly, explained by the differing definitions of ULN for AST. Moreover, previous meta-analyses have often been restricted to analysis of pre-defined biomarker thresholds whereas a strength of using IPD is that it can facilitate analysis across the continuum of biomarker thresholds. It should be noted that the original study that developed APRI and defined its thresholds was performed among active chronic hepatitis C patients in the USA[10], much older than our HBV population, with higher liver transaminases levels, and mainly Caucasian (only 8% African Americans), a very different patient population compared to African CHB patients.

The new thresholds identified in our study for APRI and GPR were particularly suitable at ruling out disease (i.e. they had high negative predictive values). The ability to rule in disease (i.e. positive predictive value) was much poorer. This phenomenon reflects a common problem with screening tests, particularly in a community setting, resulting in significant overdiagnosis when the prevalence and pre-test probability of disease is low[46]. Indeed, the reference test used in this study, TE at specific LSM thresholds, may be associated with false-positive results if liver biopsy is considered the reference standard, and particularly in a low pre-test probability population such as asymptomatic screening. One might argue that some degree of overdiagnosis of cirrhosis is acceptable since many of those with a high APRI score (in the absence of cirrhosis) have F2/F3 fibrosis or active hepatic inflammation and need antiviral therapy to prevent progression to cirrhosis or HCC, as illustrated in Fig. 5. Overdiagnosis would lead to wider use of nucleoside analogues, which are now accessible at very low cost (<40 USD per year), are generally safe and well-tolerated, and offer the benefit of pre-exposure prophylaxis for HIV, of particular importance in African countries with generalised HIV epidemics[47,48]. On the other hand, adherence to life-long antiviral therapy might be challenging and excessive overtreatment could overstretch health budgets. We aim to perform a cost-effectiveness analysis using the present data to shed light on the balance between over-treatment and widening access to care. The optimal strategy is likely to vary in different settings depending on disease prevalence and available resources.

The intended use of these biomarkers should be kept in mind when interpreting our results. Importantly, liver fibrosis assessment is not the only criterion to initiate antiviral therapy. Clinical assessment, alanine aminotransferase (ALT), age, and family history of HCC are available in most settings, and hepatitis B e-antigen (HBeAg) and HBV DNA might be available at larger centres in low- and middle-income countries. Thus, a full patient assessment may perform better than an evaluation considering APRI or GPR alone, as presented in this study. In future work we aim to optimise treatment eligibility criteria for sSA, beyond the present question of liver fibrosis assessment.

Our analysis relies on the validity of TE as a reference standard. TE is a widely used diagnostic tool, well established in high-income countries where it has largely replaced liver biopsy in routine patient care over the past decade and has been incorporated into international HBV treatment guidelines as an accurate tool to guide treatment decisions for CHB patients[5–8]. It is worth noting that TE has prognostic properties for predicting liver-related events and death, independent of its association with liver biopsy[14]. Notably, the Baveno VII 2021 consensus recommends the use of liver stiffness to define advanced

**Table 1 | Characteristics of study participants**

| Characteristic n (%) or median (IQR) | Overall | | Reason for HBV testing | | | | Missing data | |
|---|---|---|---|---|---|---|---|---|
| | | | Asymptomatic screening populations[a] | | Suspected liver disease[a] | | n (%) | |
| Total | 3548 | (100) | 2824 | (80.8) | 673 | (19.3) | | |
| Type of study | | | | | | | 0 | (0) |
| Hospital based | 2647 | (74.6) | 1923 | (68.1) | 673 | (100) | | |
| Community based | 901 | (25.4) | 901 | (31.9) | 0 | (0) | | |
| Country | | | | | | | 0 | (0) |
| Ethiopia | 1038 | (29.3) | 717 | (25.4) | 321 | (47.7) | | |
| Senegal | 868 | (24.5) | 787 | (27.9) | 75 | (11.1) | | |
| The Gambia | 797 | (22.5) | 797 | (28.2) | 0 | (0) | | |
| Zambia | 283 | (8.0) | 256 | (9.1) | 18 | (2.7) | | |
| South Africa | 240 | (6.8) | 194 | (6.9) | 45 | (6.7) | | |
| Nigeria | 190 | (5.4) | 0 | (0) | 190 | (28.2) | | |
| Malawi | 97 | (2.7) | 73 | (2.6) | 24 | (3.6) | | |
| Burkina Faso | 35 | (1.0) | | | | | | |
| Sex, male | 2133 | (60.1) | 1656 | (58.7) | 4449 | (667) | 1 | (0) |
| Age, years | 33 | (28, 41) | 33 | (28, 41) | 34 | (28, 42) | 2 | (0.1) |
| 14–29 | 1147 | (32.4) | 919 | (32.6) | 208 | (30.9) | | |
| 30–39 | 1345 | (37.9) | 1078 | (38.2) | 250 | (37.2) | | |
| 40–49 | 634 | (17.9) | 501 | (17.8) | 124 | (18.4) | | |
| ≥50 | 420 | (11.8) | 324 | (11.5) | 91 | (13.5) | | |
| Body mass index (kg/m$^2$) | 22.4 | (20.0, 25.5) | 22.5 | (20.0, 25.6) | 22.2 | (19.7, 24.7) | 426 | (12.0) |
| Overweight (25.0–29.9) | 675 | (21.6) | 536 | (21.8) | 125 | (20.5) | | |
| Obese (≥30.0) | 205 | (6.6) | 178 | (7.2) | 20 | (3.3) | | |
| Hazardous alcohol consumption[b] | 118 | (4.7) | 89 | (4.4) | 26 | (6.3) | 1,046 | (29.5) |
| HBeAg positive | 278 | (9.0) | 170 | (7.0) | 100 | (17.0) | 473 | (13.3) |
| HBV DNA (log$_{10}$ IU/ml) | 2.7 | (1.8, 3.7) | 2.6 | (1.7, 3.6) | 3.5 | (2.6, 5.9) | 576 | (16.2) |
| <2000 IU/ml | 1920 | (64.6) | 1,712 | (67.8) | 182 | (44.7) | | |
| 2000–19,999 IU/ml | 541 | (18.2) | 461 | (18.3) | 70 | (17.2) | | |
| ≥20,000 IU/ml | 511 | (17.2) | 352 | (13.9) | 155 | (38.1) | | |
| Liver stiffness measurement (kPa) | 5.5 | (4.5, 6.9) | 5.3 | (4.4, 6.6) | 6.8 | (5.0, 13.1) | 255 | (7.2) |
| ≤7.9 | 2721 | (82.6) | 2,285 | (88.5) | 395 | (59.3) | | |
| 8.0–9.5 | 203 | (6.2) | 150 | (5.8) | 52 | (7.8) | | |
| 9.6–12.2 | 128 | (3.9) | 83 | (3.2) | 43 | (6.5) | | |
| >12.2 | 241 | (7.3) | 64 | (2.5) | 176 | (26.4) | | |
| ALT (U/L) | 24 | (18, 34) | 23 | (18, 31) | 33 | (20, 49) | 44 | (1.2) |
| AST (U/L) | 28 | (22, 35) | 27 | (21, 33) | 35 | (24, 52) | 58 | (1.6) |
| GGT (U/L) | 24 | (18, 35) | 23 | (18, 33) | 30 | (18, 56) | 941 | (26.5) |
| Platelets (×10$^9$/L) | 233 | (182, 292) | 233 | (184, 292) | 230 | (170, 291) | 117 | (3.3) |
| APRI | 0.30 | (0.20, 0.46) | 0.28 | (0.19, 0.43) | 0.39 | (0.24, 0.65) | 138 | (3.9) |
| GPR | 0.17 | (0.11, 0.29) | 0.17 | (0.11, 0.27) | 0.22 | (0.12, 0.48) | 983 | (27.7) |
| FIB-4 | 0.82 | (0.56, 1.26) | 0.81 | (0.55, 1.22) | 0.88 | (0.60, 1.54) | 142 | (4.0) |

*ULN* Upper limit of normal, *IQR* interquartile range, *ALT* alanine aminotransferase, *AST* aspartate aminotransferase, *GGT* gamma-glutamyl transferase, *APRI* AST-to-platelet ratio index, *GPR* GGT-to-platelet ratio.
[a]Reason for testing for hepatitis B was missing for 51 (1.4%) of participants.
[b]Hazardous alcohol consumption was as defined by each centre; centre-specific definitions are reported in the supplementary information.

CLD, and to prognosticate risks of decompensation and liver-related death[49]. Numerous cross-sectional studies have compared TE with liver histology and found good agreement, particularly for cirrhosis. In a meta-analysis from 2016, Li et al. analysed data from 27 studies with a total of 4386 CHB patients and found an AUROC of 0.88 for F ≥ 2 and 0.93 for F4[50]. In our own study, we performed a sensitivity analysis among a subset of patients who underwent liver biopsy and found that the performance characteristics of the WHO-recommended thresholds and our newly derived rule-in and rule-out thresholds were consistent with findings from the main analysis. Indeed, liver biopsy also has its limitations due to sampling error and inter-observer variability of histological specimens, and it has been shown that AUROC > 0.90 is unachievable even for a perfect surrogate marker[51].

This study had some limitations. First, APRI and GPR displayed reduced specificity in patients with alcohol abuse. It is well-known that alcohol may cause elevations of both AST and GGT and reductions in platelet count; hence, fibrosis scores based on AST and GGT must be used with caution in patients with known alcohol misuse[52]. Second,

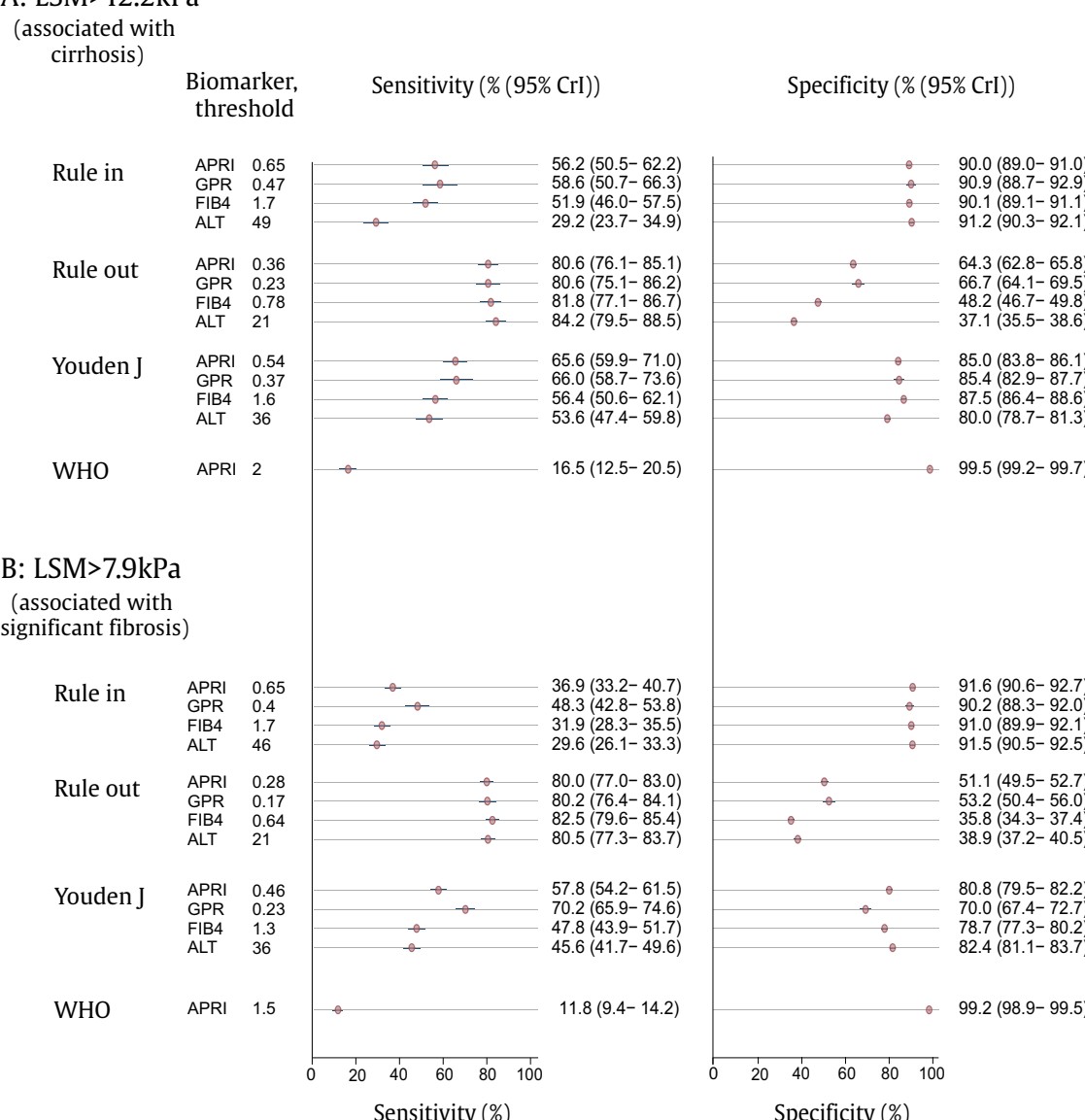

**Fig. 3 | Performance of biomarkers of liver fibrosis (APRI, GPR, FIB-4) and ALT using transient elastography as a reference for the diagnosis.** The diagnosis of **A** LSM > 12.2 kPa (associated with cirrhosis) and **B** LSM > 7.9 kPa (associated with significant fibrosis): Bayesian random effects model[a]. Point estimates are shown as red circles with 95% credible intervals shown as error bars. CrI credible interval, APRI aspartate aminotransferase-to-platelet ratio index, FIB-4 fibrosis-4 score, GPR gamma[1]glutamyl transferase-to-platelet ratio, ALT alanine aminotransferase, LSM liver stiffness measurement. [a]Bayesian bivariate random effects model (*n* = 3548 biologically independent samples) adjusted for sex, study, hazardous alcohol consumption, reason for testing (suspected liver disease or asymptomatic screening) and categorical body mass index; summary statistics show population average. Youden's J = maximisation of Youden's J statistic, with equal weight to sensitivity and specificity (J = sensitivity + specificity − 1). Rule-in models were chosen where specificity exceeds 90%; rule-out models where sensitivity >80%.

although there was an overall low risk of bias associated with the studies, a small number had non-random selection criteria or specific exclusions that could limit applicability. Third, not all participants had TE and blood tests performed the same day; however, the majority (>90%) had both tests done within the same week. Finally, TE has been associated with technical limitations including unsuccessful measurements reported in patients with ascites, obesity, or narrow intercostal spaces. We did not ascertain rates of TE test failure, and this may affect the overall applicability of our findings to the entire population with HBV.

The study had significant strengths. This is the largest and most comprehensive analysis from sSA, the most endemic region for HBV worldwide, and we had good geographical representation from East, West, and Southern Africa. We therefore believe our results are generalisable. The bivariate random effects meta-analysis (BRMA)

modelling framework we used has been widely advocated for, in diagnostic accuracy meta-analysis[53,54]. The BRMA framework accounts for both within study precision of estimates and heterogeneity between studies. The model allowed us to model the effect of patient-level covariates on diagnostic accuracy. Further, the models jointly model sensitivity and specificity, capturing the dependence between these two parameters. Finally, the Bayesian estimation framework we used allows a principled treatment of missing data.

In conclusion, APRI, at the current WHO-recommended threshold of 2.0, had poor sensitivity for the diagnosis of LSM >12.2 kPa in sSA. We developed new rule-in and rule-out thresholds for APRI and GPR, which had better discriminatory properties. The non-invasive biomarkers were better at ruling out than ruling in disease, and when applied as screening tools on a general population a certain degree of over-diagnosis must be assumed. Our data are important for informing

**Table 2 | Predictive values, sensitivities and specificities for different study populations for APRI and GPR**

| Population, diagnostic target[a] | Biomarker | Cut-off | PPV (%) | NPV (%) | Sensitivity (%) | Specificity (%) |
|---|---|---|---|---|---|---|
| Asymptomatic screening, LSM > 12.2 kPa | APRI rule-in | 0.65 | 17.0 (13.9–20.2) | 98.4 (98.0–98.8) | 56.4 (45.9–66.9) | 90.6 (89.5–91.6) |
| | GPR rule-in | 0.47 | 20.1 (16.1–24.3) | 98.5 (98.1–99.0) | 69.7 (48.0–71.4) | 91.9 (90.5–93.3) |
| | APRI Youden's J | 0.54 | 14.0 (11.9–16.0) | 98.8 (98.4–99.2) | 68.8 (58.6–78.6) | 85.6 (84.3–86.8) |
| | GPR Youden's J | 0.37 | 13.9 (11.3–16.5) | 98.6 (98.1–99.0) | 63.4 (51.9–74.5) | 86.7 (85.0–88.4) |
| | APRI rule-out | 0.36 | 7.4 (6.7–8.1) | 99.0 (98.7–99.4) | 80.7 (73.7–87.8) | 65.5 (64.0–67.1) |
| | GPR rule-out | 0.23 | 7.8 (6.8–8.8) | 98.9 (98.5–99.4) | 79.4 (69.5–88.5) | 68.2 (66.1–70.3) |
| Suspected liver disease, LSM > 12.2 kPa | APRI rule-in | 0.65 | 70.5 (65.4–75.6) | 77.3 (74.6–79.9) | 56.6 (49.9–62.8) | 86.2 (83.2–89.1) |
| | GPR rule-in | 0.47 | 74.9 (48.9–93.7) | 73.0 (68.4–77.2) | 42.5 (32.0–52.5) | 90.6 (73.9–98.7) |
| | APRI Youden's J | 0.54 | 66.0 (61.5–70.3) | 79.4 (76.6–82.3) | 64.0 (58.0–70.0) | 80.7 (77.4–84.0) |
| | GPR Youden's J | 0.37 | 72.0 (51.7–87.8) | 76.8 (72.3–81.0) | 55.2 (44.2–64.9) | 86.3 (70.0–96.3) |
| | APRI rule-out | 0.36 | 51.9 (49.0–54.7) | 83.2 (79.3–87.0) | 80.3 (75.1–85.4) | 56.7 (52.4–60.6) |
| | GPR rule-out | 0.23 | 49.7 (40.8–63.7) | 81.2 (73.5–88.6) | 80.2 (73.4–86.6) | 51.3 (34.4–73.9) |
| Asymptomatic screening, LSM > 7.9 kPa | APRI rule-in | 0.65 | 10.7 (8.5–12.9) | 97.4 (97.3–97.6) | 28.8 (23.8–33.8) | 91.8 (90.7–92.9) |
| | GPR rule-in | 0.40 | 12.2 (9.8–14.6) | 97.7 (97.5–98.0) | 38.5 (32.1–44.8) | 90.5 (89.0–92.0) |
| | APRI Youden's J | 0.46 | 8.0 (7.0–9.0) | 97.9 (97.7–98.1) | 48.8 (43.7–53.9) | 80.9 (79.4–82.3) |
| | GPR Youden's J | 0.23 | 7.0 (6.3–7.7) | 98.3 (98.0–98.6) | 63.9 (58.1–69.7) | 71.0 (68.8–73.2) |
| | APRI rule-out | 0.27 | 4.7 (4.4–5.0) | 98.5 (98.2–98.7) | 78.9 (75.0–82.9) | 45.4 (43.6–47.1) |
| | GPR rule-out | 0.17 | 5.4 (5.0–5.8) | 98.5 (98.2–98.8) | 76.5 (71.4–81.1) | 54.1 (51.7–56.4) |
| Suspected liver disease, LSM > 7.9 kPa | APRI rule-in | 0.65 | 75.1 (68.8–80.7) | 74.9 (72.9–76.9) | 47.6 (42.3–53.2) | 90.8 (88.0–93.4) |
| | GPR rule-in | 0.40 | 83.6 (69.3–93.2) | 79.6 (74.3–83.5) | 59.0 (46.0–68.4) | 93.1 (85.5–97.6) |
| | APRI Youden's J | 0.46 | 66.9 (62.5–71.5) | 81.6 (79.2–84.1) | 69.0 (64.1–73.7) | 80.1 (76.2–83.8) |
| | GPR Youden's J | 0.23 | 60.2 (48.5–70.6) | 83.4 (78.6–88.1) | 76.2 (67.7–83.5) | 69.9 (53.6–82.8) |
| | APRI rule-out | 0.27 | 44.7 (42.5–47.0) | 83.8 (79.4–88.4) | 87.9 (84.2–91.4) | 36.6 (31.8–41.8) |
| | GPR rule-out | 0.17 | 51.8 (43.8–60.3) | 84.4 (78.1–89.8) | 82.8 (74.0–90.0) | 54.5 (38.4–69.9) |

Estimates are reported as posterior means and 95% credible intervals from the Bayesian bivariate random-effects models.

[a]Cirrhosis prevalence was 2.5% among people tested for hepatitis B as part of asymptomatic screening, and 26.5% among people tested for hepatitis B due to suspected liver disease (viz with symptoms, clinical signs, or abnormal liver function tests).

clinical practice in sSA and should be considered in the next revision of the WHO hepatitis B guidelines.

## Methods

### Systematic review for relevant cohorts

We searched PubMed, Scopus, the African Index Medicus and African Journals Online for articles published until 12th July 2022 using a search strategy covering the synonyms of "hepatitis B" AND ("TE" OR "liver biopsy") AND "sSA" (Supplementary Table 7). Two investigators independently screened all identified articles and reviewed potentially eligible full-text articles for eligibility without language restrictions.

We included studies reporting on HBsAg-positive adults or adolescents (≥13 years) living in sSA who had pre-therapy LSM using TE and measurement of ALT, AST, GGT and platelet count. TE and blood tests were usually performed within the same week but an interval of up to 3 months was accepted. We excluded studies on migrants of African origin. Two investigators (AS and NR) independently extracted the variables listed in Supplementary Table 8 and evaluated the risk of bias using the QUADAS-2 assessment tool with disagreement resolved by consensus[55].

### Individual patient data

We contacted authors of all publications which met our study inclusion criteria, as well as researchers active in this area, to share IPD. For the current analysis, we excluded patients who had received anti-HBV treatment within the preceding 6 months, or who had started therapy for at least 7 days prior to the time of evaluation, patients with co-infection with hepatitis C (anti-HCV) or hepatitis D (anti-HDV) or human immunodeficiency virus (HIV), pregnant women, patients with ALT or AST exceeding five times the ULN (in accordance with European Association for the Study of the Liver [EASL] reliability recommendations for TE)[9], and those with suspected or confirmed HCC. Centre-specific definitions were used to categorise patients with hazardous alcohol consumption as described in Supplementary Table 1; the majority used the WHO AUDIT tool[56].

### Reference test

TE was performed (using the M probe) on patients who fasted for at least 2 h. The median of 10 readings was reported. The result was discarded if the IQR divided by the median exceeded 30% when the median LSM was ≥7.1 kPa[57]. We used TE thresholds of 7.9 and 12.2 kPa, associated with significant fibrosis (equivalent to METAVIR ≥ F2) and cirrhosis (F4) in cross-sectional studies, as the reference standard for assessment of the biomarkers[13,50]. In a sensitivity analysis, we explored an alternative LSM threshold associated with cirrhosis (9.5 kPa) derived from a study of 135 Gambian CHB patients who underwent liver biopsy[13].

### Index tests

We assessed the following biomarkers:

i.   APRI = [(AST (U/L)/ULN of AST) × 100]/platelet count (×10⁹/L)[10];
ii.  GPR = [GGT (U/L)/ULN of GGT × 100]/platelet count (×10⁹/L)[13];
iii. FIB-4 = (age (years) × AST (U/L))/(platelet count (×10⁹/L) × (ALT (U/L))^(1/2))[11];
iv.  ALT (U/L), as a standalone marker.

An ULN of 40 U/L for AST and 61 U/L for GGT were used in the models.

### Statistical analysis

First, we evaluated the WHO recommended rule-in thresholds for APRI (1.5 for ≥F2 and 2.0 for F4)[8]. We then optimised the rule-in

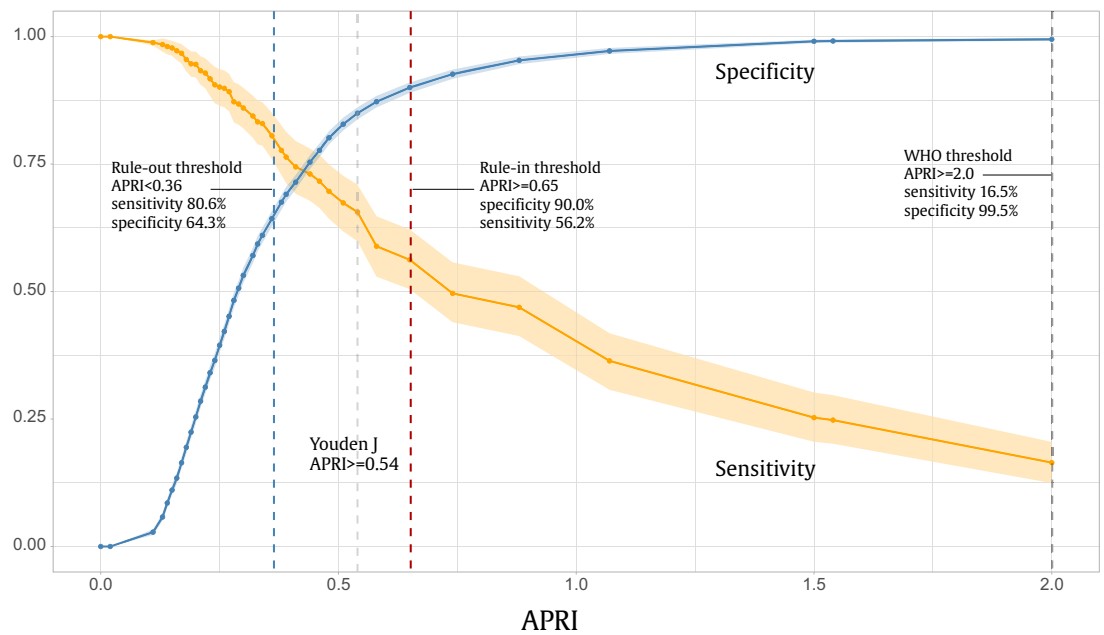

A: All participants

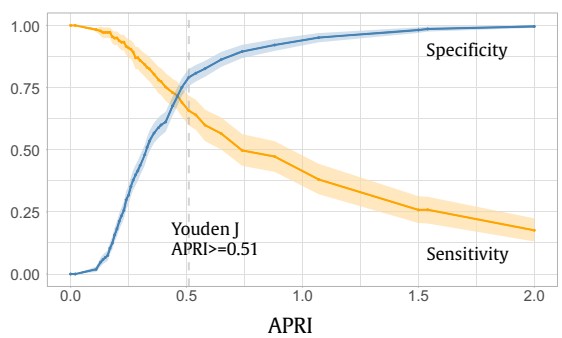

B: Asymptomatic screening

C: Suspected liver disease

**Fig. 4 | Relationship between sensitivity and specificity for APRI used to diagnose liver stiffness measurement >12.2 kPa (associated with cirrhosis).** Relationship among **A** all participants; **B** asymptomatic screening populations and **C** patients with suspected liver disease[a]. Point estimates at different APRI thresholds are shown as blue (specificity) or orange (sensitivity) dots, connected by straight line segments with 95% credible intervals shown as transparent error bands. [a]Bayesian bivariate random effects model fitted for different thresholds of APRI using 60 equally spaced quantiles. Specificity is defined as the probability of a negative test given the absence of LSM > 12.2; sensitivity is defined as the probability of a positive test, given the presence of LSM > 12.2. Source data are provided as a Source Data file.

thresholds by giving priority to specificity (≥90%), and the rule-out thresholds by giving priority to sensitivity (≥80%). Finally, we assessed thresholds defined by maximising Youden's J, equivalent to maximising the sum of sensitivity and specificity without priority to any of the two.

To calculate sensitivity and specificity, data were pooled using a single-stage IPD meta-analysis approach. We used a bivariate Bayesian random-effects meta-analysis model for sensitivity and specificity using patient-level covariates with study-level random effects to account for anticipated variability between sites (full model details are presented in Supplementary Methods 1)[54].

## Validation
We validated the model using bootstrap resampling, as widely recommended over split-sample approaches for internal validation[58]. We obtained 500 bootstrap samples from the original dataset, then fitted the model and estimated rule-in, rule-out and Youden thresholds, sensitivity and specificity at each different thresholds, and overall

AUROC. Validation results for the APRI model for cirrhosis are shown in Supplementary Methods 2. Model parameters show unimodal distributions with narrow spread indicating good stability of parameter estimates.

## Sensitivity analyses
First, we assessed the effect of using assay-specific ULN for AST, as reported by each centre. Second, we assessed the effect of using a lower liver stiffness threshold associated with cirrhosis (9.5 kPa)[13]. Third, we assessed the performance of new optimised rule-in and rule-out thresholds on a subset of patients who underwent pre-therapy liver biopsy using the METAVIR histological fibrosis scores as a reference test.

To explore variables associated with LSM thresholds indicative of cirrhosis (LSM > 12.2) and significant fibrosis (LSM > 7.9), we used mixed effects logistic regression models with study site-specific random effects by including variables that were anticipated a priori to be clinically important: age, sex, BMI category, and reason for

**Fig. 5 | Illustration of using APRI to classify patients in asymptomatic screening and suspected liver disease populations.** Using **A** rule-in and rule-out thresholds, **B** the WHO recommended threshold and **C** Youden J derived threshold[a]. [a]A single grey dot represents one person with liver stiffness measurement (LSM) < 7.9 kPa, one orange dot represents a person with LSM > 7.9 kPa associated with significant fibrosis, and one red dot represents a person with LSM > 12.2 kPa associated with cirrhosis (F4). Individual proportions rounded to nearest whole individual and represent the effect of applying APRI thresholds to the average populations included in the cohorts, stratified by reason for hepatitis B testing. Source data are provided as a Source Data file.

hepatitis B testing (asymptomatic vs. suspected CLD). To examine an association between APRI test sensitivity and LSM among patients with cirrhosis, we used the Wilcoxon rank sum test and plotted the distribution of LSM and APRI classification using a kernel density plot and a restricted cubic spline plot for LSM against test sensitivity. Analyses were conducted in R version 4.1.0 (R Foundation for Statistical Computing, Austria), JAGS 4.3.0, using the rjags package (v4.10), and Stata v17 (Statacorp, USA). All R and JAGS code is available from GitHub (https://github.com/gitMarcH/HEPSANET)[59].

## Ethical review

The study protocol was registered in PROSPERO (CRD42020218043) and was reported in accordance with the PRISMA-IPD guidelines[60].

Each participating centre obtained permission from local research ethical review committees.

## Reporting summary

Further information on research design is available in the Nature Portfolio Reporting Summary linked to this article.

## Data availability

We searched PubMed, Scopus, the African Index Medicus, and African Journals Online for relevant papers. The data that supports the findings of this study are available from the corresponding author upon reasonable request. The source data underlying Figures and Supplementary Figures are provided as a separate Source Data file. Source data are provided with this paper.

## Code availability

All the codes supporting the results of the bioinformatics analysis are available in the Github according to the link: (https://github.com/gitMarcH/HEPSANET).

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

## Acknowledgements

We would like to thank the staff at all participating centres who contributed to patient follow-and data collection. The research groups of M.L. (PROLIFICA) in The Gambia and Senegal are supported by the European Commission (FP7-HEALTH, grant ID: 265994), the Medical Research Council (MRC) in the UK (grant ID: MR/R011117/1) and Gilead Sciences (Inc., Foster City, CA, USA). A.J.S. was funded by the National Institute for Health Research (UK) with an Academic Clinical Lectureship at the University of Liverpool, UK. The statistical work was undertaken on Barkla, part of the High Performance Computing facilities at the University of Liverpool, UK. The authors assume full responsibility for analyses and interpretation of these data.

## Author contributions

A.J., A.J.S., H.D., M.S., C.W.S., M.V., Y.S. and M.L. conceptualised the study. A.J., A.J.S., E.O., M.Se, G.W., M.So, C.W.S., M.V., E.S., H.D., F.F., N.R., P.D., M.D., T.M., J.T., P.C.M., M.A., S.M., R.S., Y.S. and M.L. collected data. A.J.S. and M.Y.R.H. did the data analysis. A.J., A.J.S. and M.Y.R.H. drafted the first paper. All authors reviewed the paper and approved the final version. A.J., A.J.S. and M.Y.R.H. had access to and verified the data. A.J., A.J.S., M.Y.R.H., Y.S. and M.L. had final responsibility for the decision to submit for publication.

## Competing interests

The authors declare the following competing interests: M.L. and Y.S. have received consultancy fees and research funding from Gilead Sciences, USA. The other authors declare no competing interests.

## Additional information

[1]Department of Infectious Diseases, Vestfold Hospital, Tønsberg, Norway. [2]Institute of Clinical Medicine, University of Oslo, Oslo, Norway. [3]Department of Clinical Infection, Microbiology and Immunology, Institute of Infection, Veterinary and Ecological Sciences, University of Liverpool, Liverpool, UK. [4]Malawi-Liverpool-Wellcome Trust Clinical Research Programme, Blantyre, Malawi. [5]Department of Clinical Sciences, Liverpool School of Tropical Medicine, Liverpool, UK. [6]Faculty of Medical Sciences, University of Jos, Jos, Nigeria. [7]Service de Maladies Infectieuses et Tropicales, Centre Regional de Recherche et de Formation, Centre Hospitalier National Universitaire de Fann, Dakar, Senegal. [8]Institute of Social and Preventive Medicine, University of Bern, Bern, Switzerland. [9]Division of Hepatology, Department of Medicine, Faculty of Health Sciences, University of Cape Town, Cape Town, South Africa. [10]Department of Internal Medicine, University of Zambia, Lusaka, Zambia. [11]University of Alabama at Birmingham, Birmingham, AL, USA. [12]Medical Department, St. Paul's Hospital Millennium Medical College, Addis Ababa, Ethiopia. [13]Department of Hepatology and Gastroenterology, Hopital Principal de Dakar, Dakar, Senegal. [14]Division of Medical Virology, Stellenbosch University Faculty of Medicine and Health Sciences, Cape Town, South Africa. [15]Division of Infectious Diseases, Department of Medicine, Tygerberg Hospital and Stellenbosch University, Cape Town, South Africa. [16]Nuffield Department of Medicine, University of Oxford, Oxford, UK. [17]The Francis Crick Institute, London, UK. [18]University College London, London, UK. [19]L'Institut de Recherche en Santé, de Surveillance Épidémiologique et de Formations (IRESSEF), Dakar, Senegal. [20]Yalgado Ouédraogo University Hospital Center, Ouagadougou, Burkina Faso. [21]Unité d'Epidémiologie des Maladies Emergentes, Institut Pasteur, Paris, France. [22]Department of Metabolism, Digestion and Reproduction, Division of Digestive Diseases, Hepatology section, Imperial College London, London, UK. [23]These authors contributed equally: Asgeir Johannessen, Alexander J. Stockdale, Marc Y. R. Henrion. [24]These authors jointly supervised this work: Yusuke Shimakawa, Maud Lemoine.
✉e-mail: uxasoh@siv.no

