## [Peer review file · Nature Communications]

REVIEWER COMMENTS

Reviewer #1 (Remarks to the Author):

Thank you for your work. From this IPD meta-analysis, we learned the diagnostic accuracy of the WHO-recommended biomarker APRI and two other simple biomarkers (i.e., GPR and FIB-4) for diagnosing significant liver fibrosis and cirrhosis in sub-Saharan Africa. We are appreciated to see that APRI and GPR had the best performance than FIB-4, for the diagnosis of cirrhosis in sub-Saharan Africa. Moreover, for APRI and GPR, authors identified the optimised rule-in and rule-out thresholds for cirrhosis in sub-Saharan Africa, which may help optimise treatment guidelines in this setting.

Currently, there are multiple non-invasive methods based on inexpensive laboratory tests for predicting liver fibrosis, including APRI and FIB-4. WHO recommends APRI as the preferred non-invasive test to assess significant fibrosis or cirrhosis and FIB-4 to detect advanced fibrosis, considering lower cost, routinely available methods, and untrained staff. To date, some studies (e.g., PMID: 25132233, PMID: 31423434, ...) have so far addressed the performance of these serum biomarkers (especially APRI and FIB-4) in predicting the stages of liver fibrosis in patients with CHB.

Nevertheless, in the present study, it must be acknowledged that the strength of this study is the individual patient's data analysis. More importantly, authors identified the optimised new rule-in and rule-out thresholds for APRI and GPR for cirrhosis in sub-Saharan Africa. The study has merit for novelty and clinical impact. It is well organized and discussed. Congratulations!

There are some minor comments listed as follows:

1) We did not find the results of the assessment of heterogeneity. I^{sup} was used as a measure of heterogeneity between studies.

2) The methods were slightly confused, such as statistical analysis, maybe it could be summarized more clearly.

3) Maybe, subgroup analyses of these three non-invasive tests to diagnose significant liver fibrosis and cirrhosis could be performed, such as based on BMI and age parameters.

4) In this study, authors considered TE as a reference test for cirrhosis and significant fibrosis. As the authors state, liver stiffness measurement by TE is a widely used non-invasive diagnostic method for liver fibrosis. However, TE has some technical limitations. For example, the unsuccessful measurement in patients with obesity or ascites, which limit the application of TE in advanced liver diseases. The applicability of this technique is also limited in patients with narrow intercostal space. Maybe, several comments about TE can be added to the limitation paragraph.

Reviewer #2 (Remarks to the Author):

This study addresses the important issue of screening of fibrosis in HBV patients living low- and middle-income countries. This study is made interesting by the WHO recommendations for cirrhosis screening in underprivileged countries to propose a treatment strategy targeting patients with cirrhosis. If the WHO recommendations are kept unchanged, they will result in excluding a large number of cirrhotic patients from access to treatment due to the lack of sensitivity of the APRI score threshold of 2.

Major Comments.

1. The authors consider prevalence of the fibrosis thresholds as equivalent to the real prevalence of fibrosis and cirrhosis which appears very high because they are based on the TE thresholds. In fact, as the authors are aware these thresholds are associated with high false positive rates. It would be more appropriate to mention the prevalence of patients above the chosen TE thresholds used for screening, avoiding presenting them as thresholds for the calculation of real prevalence of cirrhosis or cirrhosis. This modification must be made throughout the manuscript because as written one might think that the real prevalence of cirrhosis and significant fibrosis in CLD patient are 40,7% and 25% s, which are clearly overestimated. In addition, median age of the study participants at 33 years makes unlikely such prevalences of fibrosis and cirrhosis.
2. In line with comment 1, the authors should discuss whether the threshold of 9.5 kPa is a relevant one when considering its low PPV for the diagnosis of cirrhosis. Although the study was published by this group, its small sample size (135) does not allow to propose a screening threshold for cirrhosis.
3. In line with comment 1, while I understand the pragmatic approach to using TE as the gold standard for fibrosis assessment, the authors should mention in the discussion that PPVs are likely overestimated when considering TE PPVs. This point is important for rigorously discussing the value of a serum screening test that has been evaluated using another physical screening method as the gold standard.
4. The authors proposed the identification of a threshold allowing a sensitivity > 80%. Can they give the statistical justification for this analysis to find an optimal threshold? In fact when looking at the results in table 2, I am not convinced that the proposed thresholds 0.36 and 0.65 are superior than the threshold defined by maximising Youden's J. Authors must provide statistical evidence that enabled them to

highlight these 2 thresholds rather than the one identified by a statistical method considered to be the most appropriate for the choice of thresholds.

5. I am confused by the presentation of the results in table 2. In fact, the authors should present the results on the overall population (CLD and asymptomatic) and then the results according to the population CLD and asymptomatic. This conduct of the analysis is also justified for the following points: a/ a sensitivity analysis can only concern a fraction of the overall population and not start with an analysis on each subpopulation; b/ in terms of screening, the approach without classification as CLD vs asymptomatic could be more efficient for massive screening; .

6. The authors should be commended for the quality of figures 4b and 5 which are informative for the reader. Conversely, figure 4a is less useful and could be passed on as an supplementary figure (similar comment for the figure 1). This would make it possible to optimize Figures 4a and 5 according to the various preceding comments (overall population, asymptomatic and CLD).

7. The authors should implement the figure 5 by better explaining the number of candidates for treatment or excluded from treatment. For example, for the rule out thresholds, explain more clearly that 660 individuals will not be treated and 340 treated if the rule out threshold is retained for therapeutic indication. Similar comments for the different thresholdsthe results including the presentation of the threshold defined by maximising Youden's J.

8. The authors should provide the results for F2 in the sensitivity analysis on patients with liver biopsy instead of claiming "Similar findings, consistent with the main analysis were observed for GPR, and for the diagnosis of significant fibrosis (F2)".

RESPONSE TO THE REVIEWERS' COMMENTS

Reviewer #1 (Remarks to the Author):

1) We did not find the results of the assessment of heterogeneity. I was used as a measure of heterogeneity between studies.

We apologize for this and have now clarified this point. The bivariate random effects model we used includes random effects for both the sensitivity and specificity components of the model. These are the usual τ^2 between-study heterogeneity parameters commonly reported in meta-analyses. Given the number of models we fitted (for different TE thresholds, biomarker types and biomarker thresholds), we previously did not report these statistics, but we agree with the reviewer that it will be helpful to report them for the main models that we describe. We have added a table listing values of τ^2 as a description of heterogeneity for sensitivity and specificity between participating sites in the various models to Appendix 15.

2) The methods were slightly confused, such as statistical analysis, maybe it could be summarized more clearly.

We thank the reviewer for this important comment. We have provided a simplified summary of the model in the section "Statistical analysis". We have amended this section to indicate that more detailed information on the model is included in the supplementary appendix (appendix 15) including a description of the mathematical parameters of the model. We have edited appendix 15 on the description of statistical methods for clarity and to provide more detail on the components of the model.

3) Maybe, subgroup analyses of these three non-invasive tests to diagnose significant liver fibrosis and cirrhosis could be performed, such as based on BMI and age parameters.

We thank the reviewer for this suggestion. The effect of subgroups can be understood with reference to the model parameters (patient-level covariates) presented in Appendix 8. We have additionally added subgroup analyses presented as a figure to illustrate the effect of these subgroups on diagnostic performance, in an additional supplementary appendix (Appendix 9).

4) In this study, authors considered TE as a reference test for cirrhosis and significant fibrosis. As the authors state, liver stiffness measurement by TE is a widely used non-invasive diagnostic method for liver fibrosis. However, TE has some technical limitations. For example, the unsuccessful measurement in patients with obesity or ascites, which limit the application of TE in advanced liver diseases. The applicability of this technique is also limited in patients with narrow intercostal space. Maybe, several comments about TE can be added to the limitation paragraph.

This is an important comment, also made by reviewer 2 and we have added these points to our discussion on limitations of the use of transient elastography (TE), as suggested. It is worth noting that the alternative to TE for assessment of liver fibrosis, a liver biopsy, is also associated with a significant range of potential limitations: spectrum bias due to lack of justification to perform biopsy in all

patients, patient refusal, and inherent problems with biopsy as a reference standard: misclassification and inter-observer variability around fibrosis interpretation. (*Fredrich-Rust et al, Nature Rev Gastro Hepatol 2016; 13(7): 402-411*).

We have excluded patients with either unreliable results as defined by Boursier et al. (*Hepatology 2013; 57(3): 1182-1191*) or invalid results. All values were collected in a fasting state and by trained operators as recommended.

TE is now recommended as a reference tool to assist decision making for commencing treatment in all international treatment guidelines for HBV including World Health Organisation (WHO), European (EASL), American (AASLD), and Asian-Pacific (APASL) hepatology societies' guidelines, and provides prognostic information about the risk of future decompensation or liver-related events.

We have added additional detail about these issues in the discussion (lines 329-332) and have provided further remarks on this matter in response to reviewer 2 below.

Reviewer #2 (Remarks to the Author):

1. The authors consider prevalence of the fibrosis thresholds as equivalent to the real prevalence of fibrosis and cirrhosis which appears very high because they are based on the TE thresholds. In fact, as the authors are aware these thresholds are associated with high false positive rates. It would be more appropriate to mention the prevalence of patients above the chosen TE thresholds used for screening, avoiding presenting them as thresholds for the calculation of real prevalence of cirrhosis or cirrhosis. This modification must be made throughout the manuscript because as written one might think that the real prevalence of cirrhosis and significant fibrosis in CLD patient are 40,7% and 25%, which are clearly overestimated. In addition, median age of the study participants at 33 years makes unlikely such prevalences of fibrosis and cirrhosis.

We agree with this important point. Any comparisons of TE thresholds with liver biopsy are subject to several issues of spectrum bias, non-applicability, the frequent lack of an intention-to-diagnose study design and substantial inter-observer variability around the liver biopsy reference standard. Regardless of its agreement with liver histology, however, TE is a strong prognostic marker of liver-related events including in patients with chronic hepatitis B (*Wong GH et al, Journal of Gastroenterology and Hepatology 2015; 30(3): 582-590*) and has gained widespread acceptance as a decision support tool for HBV treatment in international HBV treatment guidelines.

Nonetheless, we accept that any TE threshold, particularly in low-pretest probability populations such as asymptomatic screening populations, may be associated with false-positive results if liver biopsy is considered the reference standard. We agree to recharacterize the nomenclature for "cirrhosis" and "significant fibrosis" with "liver stiffness measurement >12.2 kPa" ("LSM>12.2") and "liver stiffness measurement >7.9 kPa" ("LSM>7.9") throughout the manuscript to make this point clear.

*2. In line with comment 1, the authors should discuss whether the threshold of 9.5 kPa is a relevant one when considering its low PPV for the diagnosis of cirrhosis. Although the study was published by this group (*Lemoine et al Gut 2016*), its small sample size (135) does not allow to propose a screening threshold for cirrhosis.*

We thank the reviewer for this important comment. The best TE threshold for diagnosis of fibrosis and cirrhosis remains subject of debate. We used the threshold of 12.2 kPa in the primary analysis, supported by a meta-analysis of 27 cross-sectional studies in 4,386 patients with chronic HBV in European and Asian populations (*Li et al, Aliment Pharmacol Ther 2016; 43: 458–69*). In our study, the threshold of 9.5kPa was evaluated only as a secondary exploratory sensitivity analysis since it is derived from liver biopsy data from sub-Saharan Africa.

3. In line with comment 1, while I understand the pragmatic approach to using TE as the gold standard for fibrosis assessment, the authors should mention in the discussion that PPVs are likely overestimated when considering TE PPVs. This point is important for rigorously discussing the value of a serum screening test that has been evaluated using another physical screening method as the gold standard.

We have revised our discussion to emphasise this important point (line 280-284).

4. The authors proposed the identification of a threshold allowing a sensitivity > 80%. Can they give the statistical justification for this analysis to find an optimal threshold? In fact when looking at the results in table 2, I am not convinced that the proposed thresholds 0.36 and 0.65 are superior than the threshold defined by maximising Youden's J. Authors must provide statistical evidence that enabled them to highlight these 2 thresholds rather than the one identified by a statistical method considered to be the most appropriate for the choice of thresholds.

We thank the reviewer for raising this issue. Significant trade-offs between maximising sensitivity and specificity must be made when faced with the need to both avoid missed opportunities to treat HBV-associated liver disease and concerns about over-treatment in the context of limited resources. These are policy questions.

We have therefore chosen to present three choices over thresholds that could be reasonably selected by policy makers or clinicians' preference depending on whether sensitivity (minimising under-treatment) or specificity (avoiding over-treatment) is prioritised. The choices of 90% specificity and 80% sensitivity were chosen a priori by consensus of the study team as reasonable choices for policymakers (cf. similar studies: *FE Mozes et al, Gut 2021; MJ Sonneveld et al, Lancet Gastro Hepatol 2019*). Because the effect of continuous APRI threshold choices on sensitivity and specificity are shown in figure 4, in fact any APRI threshold may be selected and the consequences on these diagnostic parameters can be visualised. The mathematically derived Youden's J, which does not take clinical/policy priorities into account, is additionally presented.

5. I am confused by the presentation of the results in table 2. In fact, the authors should present the results on the overall population (CLD and asymptomatic) and then the results according to the population CLD and asymptomatic. This conduct of the analysis is also justified for the following points: a/ a sensitivity analysis can only concern a fraction of the overall population and not start with an analysis on each subpopulation; b/ in terms of screening, the approach without classification as CLD vs asymptomatic could be more efficient for massive screening; .

We thank the reviewer for this comment and apologise for the confusion. The intention of table 2 is to present the predictive values (PPV and NPV) in relation to the pre-test probabilities associated with the population type. It is not a sensitivity analysis but applies to the entire study population. The population type will always be apparent to clinicians or policymakers: a massive screening approach would be consistent with “asymptomatic screening”, whereas patients presenting or referred to a physician with suspected features of chronic liver disease would fulfil the category of “suspected liver disease”. We are concerned that providing the predictive values for the entire population to table 2 would merely arbitrarily reflect the relative composition of the included research studies (for example the proportion of research conducted in community vs hospital settings) without presenting additional informative data. Therefore, we propose to keep it unchanged.

6. The authors should be commended for the quality of figures 4b and 5 which are informative for the reader. Conversely, figure 4a is less useful and could be passed on as an supplementary figure (similar comment for the figure 1). This would make it possible to optimize Figures 4a and 5 according to the various preceding comments (overall population, asymptomatic and CLD).

We have moved Figure 4a to the supplementary appendix as requested. We assume the reviewer means to recommend optimising figure 4b for the three different populations described. We have provided the combined sensitivity/specificity figure for the two subgroups as requested within a revised Figure 4. Figure 5 illustrates the effect of pre-test probability on predictive values (PPV and NPV) within the two types of populations (asymptomatic and CLD). As per comment 5 above, Figure 5 is only interpretable in the context of a specified pre-test probability. Therefore, adding the entire study population to figure 5 would arbitrarily reflect the composition of the included research (the relative mixture of asymptomatic and liver disease populations), rather than provide additional insight into predictive values. We would prefer to retain figure 1 as it clearly demonstrates the geographic composition and sample size of included studies, which is of value in interpreting our study.

7. The authors should implement the figure 5 by better explaining the number of candidates for treatment or excluded from treatment. For example, for the rule out thresholds, explain more clearly that 660 individuals will not be treated and 340 treated if the rule out threshold is retained for therapeutic indication. Similar comments for the different thresholdsthe results including the presentation of the threshold defined by maximising Youden's J.

Thanks for pointing this out. We have now revised figure 5 for clarity by showing that the overall population is divided by the APRI thresholds, using arrows to indicate the division of the population into APRI threshold categories.

8. The authors should provide the results for F2 in the sensitivity analysis on patients with liver biopsy instead of claiming “Similar findings, consistent with the main analysis were observed for GPR, and for the diagnosis of significant fibrosis (F2)”.

The additional results for F2 and for GPR are provided in Appendix 11.

REVIEWERS' COMMENTS

Reviewer #2 (Remarks to the Author):

The revised manuscript has been significantly improved.

I do not have additional comments.